# A Blended Finance Framework for Heritage-Led Urban Regeneration

**Bonnie Burnham**

Cultural Heritage Finance Alliance, New York, NY 10023, USA; bburnham@heritagefinance.org;
Tel.: +1-917-366-4554

**Abstract:** The inclusion of heritage conservation in the United Nations Sustainable Development Goals for 2030, target 11.4, stimulated a broad dialogue among heritage conservation practitioners intent on framing a meaningful role for heritage assets in historic built environments as contributors to sustainable development. Heritage-led regeneration positively impacts many aspects of society, community life, and the public realm, and can also play an important role in reaching zero-carbon environmental conservation goals by slowing the extraction of natural resources for construction, reducing the quantity of building materials sent to landfills, and using traditional technologies and knowledge to reduce operational energy use. Heritage regeneration can also be a strong contributor to economic growth, as restored and reused properties create wealth, serve as community social magnets, and attract prestige and visitors. However, there is little progress towards positioning heritage conservation as a focal point for multilateral public-private co-financing projects and partnerships. In 2021, the Cultural Heritage Finance Alliance (CHiFA) published research about successful models of urban heritage regeneration that engage public-private cooperation. CHiFA now presents a process, developed as part of a study commissioned by the Inter-American Development Bank (IDB), for advancing projects that maximize investment in heritage-led urban regeneration, matching financing strategies with local opportunities, legal frameworks, enabling tools, and the requirements of prospective investors. The result is a marketplace and ecosystem that support civic and community interests through long-term, multi-party collaboration using blended capital investment in heritage as a sustainable development strategy.

**Keywords:** heritage finance; blended finance; sustainable development; urban regeneration

## 1. Introduction

> *"Culture is key to what makes cities attractive, creative and sustainable. History shows that culture is at the heart of urban development, evidenced through cultural landmarks, heritage and traditions. Without culture, cities as vibrant life-spaces do not exist; they are merely concrete and steel constructions, prone to social degradation and fracture. It is culture that makes the difference. How can culture be integrated into urban strategies to ensure their sustainability?"* [1]

This statement, in the introduction to the book Culture Urban Future, published by UNESCO in 2016, set the stage within the field of heritage conservation for a wide reconsideration of how it contributes to planetary sustainability and the future of the built environment. Its publication coincided with the inclusion of heritage conservation in the United Nations Sustainable 2030 Development Goals (target 11.4) [2] and in the New Urban Agenda of UN-HABITAT III [3]. The statement speaks to the value we traditionally place on culture as the cohesive, unifying element that holds our communities together through a sense of identity. Today however, culture and heritage have a much larger role to play in the economic life of communities, the control of climate change, and the advancement of social equity that are at the forefront of our concerns.

Urban heritage, including tangible and intangible elements, is key to the character of a city or urban area, and comprises one of its principal values. The importance of urban heritage conservation is underscored by the fact that, as of 2019, more than 70% of cultural properties inscribed on the World Heritage List were located in or encompass urban areas, including more than 2,700 cities and towns in 624 out of a total 1092 inscribed properties [4]. Heritage is an important part of the planetary legacy that is presently under threat due to a range of factors including unmanaged and inequitable growth, economic decline in city centers, and the effects of climate change. Failure to integrate these assets into changing urban environments results in irreparable losses to future generations [5].

To integrate concern for the conservation of traditional urban environments into the sustainability agenda, in 2011 the World Heritage Centre developed Historic Urban Landscape (HUL) recommendations for the proper management of change impacting heritage assets. The HUL recommendations call for regulatory systems and financial tools "aimed at building capacities and supporting innovate income-generating development, rooted in tradition [6]". The most vigorous response to these Recommendations to date comes from the developed world. European policy initiatives in the last few years, such as the 2018 Davos Declaration [7] and the European Union Horizon 2020 program, focused on heritage as a key element in supporting the region's sense of its core values. Under Horizon 2020, the recently completed EU CLIC program (Circular models Leveraging Investments in Cultural heritage adaptive reuse, i.e.,"ARCH") produced dozens of research papers as well as demonstration projects. Dr. Tracy Pickerill of Technical University Dublin contributed an overview of financial and nonfinancial instruments and investment leverage enablers for cultural heritage adaptive reuse. The paper presents a toolkit to help urban authorities understand how to create a blended pool of capital and leverage enablers. It also collects dozens of evidence-based examples of collaborative heritage revitalization financing initiatives at all scales [8].

Another CLIC report, by professors Gillian Foster of the Vienna University of Economics and Business and Ruba Saleh of the ICHEC Brussels Management School, develops a cultural heritage index to measure investment opportunities in adaptive reuse across European cities. This index applies 15 indicators in three dimensions (cultural stock, environmental stewardship, and socioeconomic factors) to create a composite indicator, and then evaluates 190 European cities against it. Meaningful for the study is the outcome: the top 20 indexed cities were within seven Western European countries. Ninety percent have at least one World Heritage site within their perimeter, and many are European capitals and hence, historical centers of wealth and culture. The article also presents a compelling summation of how Adaptive Reuse of Cultural Heritage (ARCH) contributes to circular development, as follows

Adaptive Reuse of Cultural Heritage:

- Extends the lifespan of existing buildings and slows the extraction of natural resources and energy use for new buildings;
- Capitalizes on local authenticity;
- Supports legal protections that often come with governmental financial commitments and incentives;
- Preserves cultural, emotional, social, and physical values.

Reaching sustainability goals in areas with rich architectural heritage will be difficult without Adaptive Reuse of Cultural Heritage (ARCH) [9].

The extension of this research and indexing to other parts of the world will provide an opportunity to evaluate the readiness of a city to embark upon a heritage regeneration initiative and to commit the personnel and resources needed to implement such a program. This may, however, be beyond the reach of many developing countries. In addition, these discussions take place largely within the heritage conservation sector itself, and there is little progress to date towards integrated coordination between government agencies or the organization of multilateral public-private co-financing and blended capital projects as a strategy for sustainable development.

Across the world, cities are resetting their agendas in response to the global pandemic. Many are centering their strategies on culture, especially in the developed world. [10]. For the heritage conservation field, this offers an opportunity as well as a challenge. Organizations active in the field are engaging in discussions about a new paradigm, one that reduces the top-down decision-making and financing of preservation work, and favors the inclusion of society, community, and private finance in heritage stewardship. However, there are still very few collaborative initiatives employing pooled public and private resources that bring together urban planning, natural resource conservation, heritage, community development, and finance.

## 2. Materials and Methods: A Study on Heritage Financing Models, from State-Sponsored to Privately Financed

The Cultural Heritage Finance Alliance (CHiFA), an organization that promotes heritage-led regeneration through collaborative and innovative financing solutions, conducted research in 2020 about successful models of urban heritage regeneration. This research was published in two reports: a paper entitled Impact and Identity, Investing in Heritage for Sustainable Development [11] and a more detailed compilation entitled Case Studies in Urban Regeneration [12]. Its publications present a survey of models for heritage-led regeneration, ranging from public sponsorship to private initiative, and propose a process to bridge gaps between public and private finance and develop collaborative initiatives.

The research focused on six case studies across the world of successful heritage-led regeneration initiatives involving nongovernmental players. Its conclusions are based on an analysis of published materials and interviews conducted with principals behind the work to learn how success was achieved for each model, and on what scale. The case studies reveal a spectrum of financial and management strategies, ranging from projects orchestrated by governments and utilizing debt as the financial vehicle (Fez and UK) to those that were entirely entrepreneurial with private financing (Panama City and Yangon). They also include two examples of private-led initiatives that take advantage of governmental incentives and complementary commitments to private investment, drawing upon both philanthropic and investment capital (Mexico City and Amsterdam). Each example has its own strengths and weaknesses, summarized in Table 1, below. A description of the characteristics of each model follows.

**Table 1.** Comparative advantages of different financing instruments.

| Financing Instrument | Strengths | Weaknesses | Requirements |
|---|---|---|---|
| Development Loan | Large-scale financing at low cost; Feasibility vetted and monitored by lending agency; Highly transparent | Investments only in public properties; External participants not part of deal structure | Government approval of scope and payback of borrowed funds; Heritage may not be a priority for public borrowers |
| Revolving Fund | Evergreen funding recycled to numerous projects; Capacity building for clientele; Helps facilitate urban strategies; Aligns with other government funding; May build a bridge to private finance | Scale depends on size of marketplace, usually small; Model has been confined largely to developed world; Subsidy may be needed for operating costs | Government capital contribution to launch; Independence of decision-making during operation |
| Public-Private Cooperation | Excellent opportunity for blended financing; Designed interaction between sectors results in strong local transformation; Collaboration with enablers and multi-sector institutions | Requires long-term commitment to cooperative management structure from all participating parties | Strong government backing and private-sector leadership; Large-scale urban vision and strong coordination in its realization |
| Entrepreneurial Leadership | Flexible choices of projects and scopes of work; Accountability to investors; Cost-effective and efficient | Not generally integrated with public sector priorities and other forms of capital; Success may depend on incentives or complementary philanthropic support | Operators need solid expertise in technical and financial project management |

## 2.1. The Intergovernmental Finance Model

Intergovernmental financing is a key tool for urban regeneration in developing countries. The principal goal is poverty reduction. Intergovernmental funding is directed to heritage regeneration in cities with architectural heritage of high significance where urban decline resulted in substandard inner-city living conditions. The financing, in the form of low-cost loans, is intended to result in an improvement in these conditions, as well as an economic stimulus from which the whole community stands to benefit. The lending package may be large by heritage funding standards, and the term of repayment long. Through improvement of the infrastructure supporting the historic environment, and incorporation of pollution-reduction and sanitation measures, improved transportation, and other public services, these programs contribute materially to the quality of urban life. They set the stage for additional nongovernmental investment; however, there is rarely a conscious effort to engage with partners outside the governmental institutions that are the lenders' clients.

CHiFA studied the World Bank-Moroccan government financing for the Medina of Fez through a loan program launched in 1998 and closed in 2006. The project was a banner initiative of the World Bank at a time when, under President James Wolfensohn, it focused on culture as an economic building block; the centerpiece of its Millennial program [13]. A city-funded nonprofit organization, ADER-Fez, was created to implement the project, and currently coordinates heritage regeneration projects in the city and manages other medina renewal initiatives across Morocco.

Fez became the model for a national program for the renovation of Moroccan historic medinas. Residents were the focus of the plan, as many houses in the medina were in a state of collapse, and water and airborne pollution were health threats. The program produced decongestion in the city center, and facilitated opening tourist routes which contributed greatly to the development of a tourism economy in the city. Automobile traffic was reduced drastically or eliminated.

The loan package foresaw considerable investment by the private sector through the improvement of privately owned properties and investment in new facilities mandated by the program, such as parking garages. This investment was slow to materialize, and initially the program fell far short of its ambitious financial targets. However, in the subsequent decade, private investment grew, amounting to three or four times the USD 14 million loan package. Over time, as the Moroccan government continued to strongly promote and make additional investments in the medina, Fez became an attractive destination for international tourism. However, with no formal structure at the outset to attract complementary funding from a range of sources, the results unfolded over a decade after the completion of the catalytic loan [14].

## 2.2. The Revolving Loan Fund: Evergreen Funding for Heritage Property Owners

Revolving loan funds are a widely used tool in the heritage conservation field, especially in the developed world, to encourage investment in governmentally protected heritage properties that are not publicly owned. Typically, funds are allocated by a government agency and handed off to a nonprofit organization—one that is created for the purpose of operating the fund, or is already active in local heritage preservation. Loans are generally offered at below-market rates to provide capital that may not be readily available from commercial institutions. Revolving funds are also used by local nonprofit organizations in many communities as bridge funding to purchase at-risk properties, which are then repositioned for resale with preservation covenants attached. To date, most heritage revolving funds are too small to finance large-scale projects or to fully cover their operational costs, and are thus operated by an existing preservation organization that uses this vehicle to supplement other programs, or supported by ongoing governmental subsidies.

CHiFA's case study focused on the Architectural Heritage Fund (AHF), active in the UK since 1976. AHF operates as a charity, but receives periodic renewals of public funds to help cover its important capacity-building activities through consulting services offered to

its clientele. AHT also raises funds from foundations and other charitable sources, and as of 2019, operates an impact fund with investor-partners. AHT's clientele consists of Building Preservation Trusts (BPTs), which are local organizations formed to renovate and reuse specific properties, and other nonprofit organizations seeking to reuse historic structures for social benefit. AHT's flexible financing and program design allow for alignment with other governmental funding programs, such as High Street improvement districts, which focus on reversing urban blight, and the Heritage Lottery Fund (HLF). Nearly half of the projects financed by AHT go on to receive HLF funding. In addition, AHT provides matching funds to incentivize local Community Shares, a national initiative that encourages private individuals to invest in local community assets. The Fund's programs are tailored to catalyze other forms of government support, offering early-stage grants, short-term loans for project preparation, and larger financing for major capital projects, often on a matching basis. With its current GBP 17.5 million in working capital, the footprint is impressive. In 45 years, AHT invested GBP 125 million in more than 600 projects across the UK. At its present scale, the operation is not financially self-sustaining, given the extensive involvement of its staff in shaping incoming projects with partners. Mainstay operational support from the government covers the gap [15].

*2.3. Aligned Public-Private Interests*

The third model considers an integrated program between public and private partners through a coordination of agendas and outcomes. Public Private Partnerships (PPPs) may be designed to coordinate such complementary work. However, CHiFA finds that theses formal partnership structures, limited to specific contractual projects, often cannot address the full scope of a community's needs and aspirations for regenerating public life. They are one-off and require continuous ongoing governmental monitoring over the life of the PPP contract. A more informal framework, established under private leadership, but enabled by public commitments of funding and facilitation, is a more comprehensive and dynamic working model; this model can be developed, in principle, in any community seeking to bring about transformative change by engaging blended capital. Enabling vehicles, created by the government and ranging from bonds and tax incentives to property concessions and direct public expenditure, are designed to attract private financing. By casting a wide net and providing a framework for interaction, this collaborative engagement can lead to a long-term commitment of shared responsibility and dialogue, with strategies and new priorities reset over time through community consultation.

The historic center of Mexico City was chosen as a case study in this form of coordinated intervention, leveraging capital from a range of sources for common benefit. The decade-long initiative was launched in 2000 under former Mexican President Vicente Fox and Mexico City Mayor Manuel Lopez-Obrador, and led by the country's most successful businessman, Carlos Slim Helú. The goal was to reverse the decline in commercial, residential, and public life (and public safety) in the city's historic district, which over the prior 50 years experienced steep decline. A coordinated initiative followed city-determined priority districts, installing new infrastructure, bus and bicycle lanes, pedestrian zones and shop arcades for vendors who had occupied streets formerly choked with traffic. Public parks and buildings were restored by the national government and new lighting was installed. All investments—municipal in the form of infrastructure; national through government renovation of monuments and public spaces, as well as through social programs; and private through real estate investments and philanthropic contributions—were coordinated through a high-level planning committee comprising representatives of public agencies, nongovernmental and academic institutions, intellectual and artistic communities, and the business sector.

The succeeding Mayor, Marcelo Ebrard (term: 2006–2012), inherited an increasingly dynamic situation and radically expanded the incentives for private investment; as a result, a vast number of new new projects were adopted by other companies. Overall, between 2001 and 2012, private investment in the historic center totaled USD 8 billion, with the

private sector far outpacing public investment of USD 193M. Multiyear plans are now renewed on six-year cycles to coincide with mayoral terms as the regeneration program pushes beyond the boundaries of the initial scheme. The city center's population grew to a sTable 40,000 in 2019 (pre pandemic) from a nadir of 5000 in 2000; the area has a vi brant public life. The creation of many new white- and blue-collar jobs in technology, commerce, and hospitality avoided the negative impacts of gentrification. This successful transformation can be considered as a regional, if not global, model [16].

*2.4. Entrepreneurial Investment for Public Good*

The three models described above deploy different financing strategies led by the public sector to produce results including conservation, social integration, economic growth, and improvement in public life. However, not all successful urban regeneration initiatives are the outcome of government leadership. Many are social enterprises generated by civic interest and entrepreneurship. Given the complexities of participating in heritage conservation as a profit-making activity, investors in this sector are looking for a convergence in a range of market factors: depressed property value coupled with the potential for short-term gain in the property's ability to generate income; enabling factors that help facilitate the process; and the availability of incentives that help offset the cost of the investment. In addition, almost all heritage investors are expecting intangible value from the investment through the prestige of association with exceptional cultural resources; through the improvement of quality of life in their chosen environment, or through the accomplishment of specific social objectives. Financing vehicles are often innovative, combining debt and equity instruments, tax incentives and subsidies, land use concessions, and property development rights transfers. The corporate framework created for these investments is tailored to a project's structure, anticipated outcomes, and associated legal restrictions or incentives. The investor must also have significant technical and financial capacities to implement projects that meet regulatory standards for protected buildings and sites, as well as compliance standards for new carbon reduction requirements.

CHiFA case studies focused on three private-sector investors in vastly different geographical and economic environments to learn how flexible investment strategies adapt to changing conditions over time: the Stadsherstel in Amsterdam, the Conservatorio S.A. in Panama City, and the Doh Eain in Yangon. A fourth case examined for its corporate structure is the private investment initiative, the Sociedad Inmobiliario Centro Histórico and its sister Fundación Histórico, in Mexico City, a hybrid with for-profit and nonprofit arms. Working with different approaches and at different scales, they address similar community development goals and achieved positive outcomes while delivering profits to investors.

Stadsherstel Amsterdam (Society for City Recovery) was founded in the 1950s when the city was in post-war decline and faced the risk of loss of character through adverse urban renewal in its historic center. A civic group formed a housing association to purchase threatened canal houses, anchoring neighborhoods through strategic building purchases and providing stable housing for existing residents. Dividends to its investors, mainly Dutch banks and insurance companies, were capped at 5%. Over time, the company's footprint grew, and the municipality became a minor investor without taking a significant role in decision-making. Strategically acquiring property over a period of decades, it had a profound effect on the city's sustainable growth by ensuring the preservation of its scale. In recent years, the company modified its structure to incorporate nonprofit activity and the management of nonresidential buildings and cultural edifices within its portfolio. The model was replicated throughout Holland and in some other cities around the world [17].

The Mexican Sociedad Inmobilario del Centro Histórico, and its sister nonprofit Fundación Centro Histórico, became active in 2001. The business arm offered its investors no guaranteed return investment; however, at the end of five years investors had the option to cash out, with a guarantee to repay their investment, or become stockholders. All opted to stay, and the company still owns its properties (as does Stadsherstel), which have

more than tripled in value. From the outset, philanthropic and corporate contributions, in addition to investments, were made by the principals through the Fundación Centro Histórico to support cultural and social programs in the city.

In Panama, Conservatorio, S.A., which began operations in 2005, set its agenda from the outset to address community social objectives as an outcome of its projects, and recently earned accreditation as a B-Corporation. It offers both debt and equity investments, the latter at market rates of return, and attracted impact-minded investors to its housing and mixed use projects. The company supports several local charitable organizations, including one that rehabilitates former gang members through job training, and includes affordable housing in all its residential projects [18].

All of the above companies envision long-term ownership of the properties they renovate. In contrast, Doh Eain, based in Yangon, works with residential property owners in the historic center who have little access to capital, to convert their properties into revenue-generating assets by taking over their management and use for a fixed period of time (typically 8 years), carrying out renovations at their own cost and returning the property to the owners at the end of the term. Doh Eain also works with both debt and equity investors. It makes complementary philanthropic investments in public space surrounding its buildings in a city that has few public amenities and struggles with pollution, traffic congestion, and overcrowding. Since the 2020 coup in Myanmar, Doh Eain was able to continue its operations in Yangon, but is expanding its operational model into other cities with deteriorating historic urban cores and limited capital markets for improvements [19].

### 2.5. Case Study Conclusions

Taken together, these six case studies validate the pivotal role heritage conservation can play in transforming historic city centers as a strategy for economic regeneration that reinforces community bonds and traditions, builds wealth, and provides a host of social and environmental benefits. By examining each approach individually, relative strengths and weaknesses emerge for each financing strategy that help determine under which circumstances the different financing strategies will be effective. The development loan helps strengthen public stewardship of key heritage assets, and may help stimulate complementary private investment. The revolving fund allows local organizations to take critical first steps in planning and orchestrating preservation work and creates a pathway to other government funding. Public-private cooperation balances the contributions of each sector to maximize results, but requires a framework of close coordination. Privately led initiatives demonstrate the vitality, ingenuity, and range of market-driven investment strategies, independent from public-sector leadership. (See Table 1, above.)

### 2.6. Incentivization and Risk Mitigation

To compete for private capital, heritage conservation projects need to meet investor requirements concerning risk reduction and proof of positive impact. To reach potential scale, the public sector may need to provide incentives to prospective investors. These can include tax benefits, investment guarantees, and subsidies for certain kinds of uses, such as affordable housing, that provide community benefits. Working with all these enabling devices, the investor may be able to reduce financial exposure to zero.

High-level political backing is a critical ingredient for creating an environment that encourages nongovernmental investment. Government agencies responsible for approving projects will need to expedite permits so that projects can proceed on schedule and meet deadlines, avoiding costly delays. Investors interviewed by CHiFA emphasize that government delays in processing approvals are the greatest investment risk in the rehabilitation of heritage properties. Providing coordination through a one-stop shop, or "open window" in the office of the regulatory authority responsible for approvals, is the single most important device to encourage independent, profit-driven engagement.

### 3. Results: A Four-Step Process for Attracting Private Investment

On completion of the CHiFA case study research, a contract with the Inter-American Development Bank (IDB) provided an opportunity to expand upon the lessons learned by creating a step-by-step process for municipalities to follow in order to engage public and private support in urban regeneration. The Living Heritage (*Patrimonio Vivo*) program of IDB was launched in 2019 [20]. It supports urban planning in Latin American and Caribbean cities centered on tangible and intangible heritage resources with the goal of promoting their conservation and enhancement as a means to achieve sustainable urban development [21]. Urban plans designed under the Living Heritage program's sponsorship are positioned to attract support through the IDB's main loan financing programs.

In 2021, Living Heritage sponsored a study to set forth a methodology for public-private co-engagement in heritage regeneration. The result will be a template for developing public-private cooperation in heritage-led regeneration. The process matches financing strategies with local opportunities, legal frameworks, and enabling tools. An extension of established urban and conservation planning methodologies, it focuses an economic lens on the question of how to take advantage of financing opportunities in the local environment that are rarely accessed in the heritage conservation field.

The report, to be published by IDB in 2022, lays out a four-step process that articulates a vision and strategy for public-private cooperation; applies this strategy to the historic urban area in order to identify the potential scale of the undertaking and highlight opportunities, obstacles and risks; assesses available financial and enabling instruments and community partners; and details a management structure for collaborative engagement. This exercise sets the stage for the implementation of pilot projects adopted by nongovernmental partners to complement IDB's public-sector commitments [22].

#### 3.1. Step 1: Vision and Strategy

A feasibility study sets a vision and strategy to frame the goals of collaborative engagement. This step gathers information on local demographics, economic and physical conditions, historic property ownership parameters, legal frameworks and constraints, incentives and governmental priorities. This research provides a snapshot of where opportunities may lie. It identifies local partnerships that may be productive. A SWOT analysis identifies obstacles to be overcome to successfully engage private and other nongovernmental finance. Criteria for selection of privately financed pilot projects are set and tested through a small group of case studies.

#### 3.2. Step 2: Framework Plan

The conclusions of the feasibility study are applied to the entire historic urban area to envision how a pilot program can be brought to scale; how it is focused; and how public investments in infrastructure, public space, and conservation of monuments will catalyze private investment in specific geographically related projects, as well as the development of opportunities for community services and benefits to be built into project deliverables.

#### 3.3. Step 3: Identifying a Funding Pool

From amongst dozens of potential financing and enabling instruments that can be used to capitalize sustainable development, those appropriate to the specific environment and project are selected; a business plan, including cost projections, places the proposed interventions within a financial offering, which may include the delivery of impact goals as well as financial return on investment.

#### 3.4. Step 4: Management Framework

With a strategy, a high-level vision, specific pilot projects selected for their impact and replicability, a cost assessment in hand, and a business plan with precise project delivery and financial return expectations projected across the life of the project, the management framework becomes the critical success factor for a heritage regeneration model's delivery.

Diverse responsibilities for project management; fund management; and communication with stakeholders, investors, and regulatory bodies requires a competent, transparent, and efficient management entity. The nature and legal character of the management agency may vary according to specific local circumstances and needs.

The process is intended to bring together projects that create synergies within a geographical context and further catalyze other opportunities by calling attention to the potential scale and impact of the opportunity at hand. A management structure is created that ensures transparency while stimulating a wide public conversation amongst stakeholders and exposing the opportunity to a wide range of potential financial participants from different strata and sectors of the marketplace (see Figure 1 below).

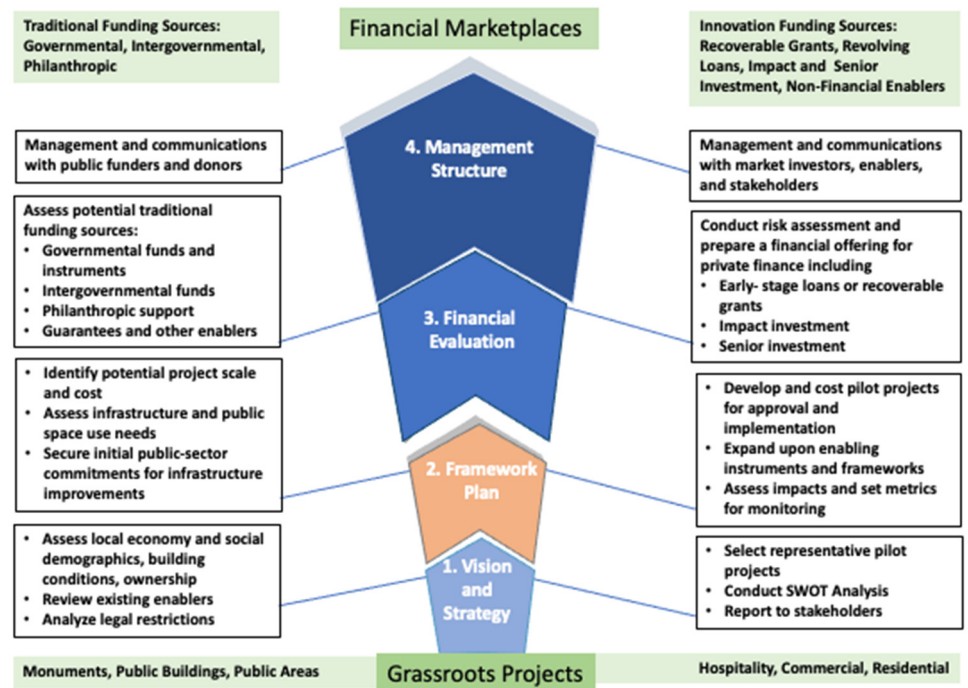

**Figure 1.** Graphic depiction of the four-step process for creation of a blended finance strategy.

## 4. Discussion: Evaluating the Viability of the Working Model

The goal of heritage-based regeneration is to use existing architectural resources with intrinsic economic and community value to stimulate sustainable local growth that is harmonious with the community's historic traditions and self-image. This process results in the growth of an ecosystem with civic interests, social support structures, regulatory protections, and innovation finance working in harmony. The development of this ecosystem results in manageable growth. The outcome of a successful urban regeneration initiative is a dynamic and potentially long-term and self-renewing partnership between diverse players, each bringing their capacities, resources, and constituencies to a collaboration that leverages the contributions of each participant and group for the benefit of all.

Success depends on key factors that the case study analysis revealed:

- Political will and charismatic local leadership;
- Capable and empowered local partners;
- Risk mitigation and investment incentives;
- Measurable social and environmental impacts;
- Transparent and efficient management structure.

Today, public policies protecting heritage assets, and the mechanics developed by the public sector to implement these policies, often stand in the way of investment. Investors shy away from layers of regulation and a long-term horizon for return on capital. If this marketplace is to grow, compelling case studies with powerful financial outcomes need

to be more widely communicated. The ecosystem needs patient capital and financial intermediaries who can deliver it. Above all, political will from the top of the system may be the most important element of success. Revolving funds can catalyze projects and bring them to the point of investability, but then long-term investments are needed. There are many prospective sources. Institutional investors, donor-advised funds, program-related investments by foundations, and impact investors looking for ESG returns all represent strong capital marketplace potential; but their engagement with heritage is relatively new. The four-step process proposed above, which requires an initial investment for its own development, is intended to bring a given initiative to the point of market readiness. At this point, intermediaries skilled in designing and marketing sophisticated financial transactions are needed to bring the product to its intended investor audiences.

In other sectors, such as nature conservation, there are numerous intermediaries and green banks able to structure large- and small-scale transactions. The nonprofit Coalition for Investment in Conservation, operated under the umbrella of the International Union for the Conservation of Nature (an intergovernmental UNESCO affiliate) documents models for expanding investment in nature conservation work through "blueprints," which outline a strategy for approaching a specific conservation issue, such as forest management, fisheries management, or sustainable agriculture [23]. As heritage conservation is not yet well structured for investment, the field presently lacks such intermediaries except in specialized markets, such as the marketplace created by national Historic Preservation Tax Credits in the United States [24].

### 4.1. Impact Measurement

Impact measurement is important to attract a new category of investors interested in supporting sustainability. Heritage conservation stands at a unique nexus at the intersection of many community, social, environmental, and cultural interests. These positive community impacts must be translated into concrete goals and deliverables on a project-by-project basis. This practice is new to the field, and one holding great potential for expanding the heritage investment pool to include impact investors.

Laws around the world are evolving toward net zero carbon in construction and the built environment. Buildings account for nearly 40% of carbon emissions today. Building industry carbon emissions take three forms: embodied carbon (in the construction materials), operational carbon, and final energy in landfill emissions. Existing buildings have an advantage in all three categories: reduction in the use of new materials by saving existing ones; traditional technologies for operating existing buildings, such as thicker walls and higher ceilings, more resilient materials, energy efficient elevations to capture light and heat, and construction based on traditional knowledge; and the reuse of existing structures, which avoids sending vast amounts of demolition materials to landfills.

Capitalizing on the opportunity of the 2021 Paris environmental accords, numerous heritage organizations, such as the Institute of Historic Building Conservation (which issued daily podcasts during the climate conference and the Climate Heritage Network (CHN) (which issued a manifesto and announced a Race to Resilience at Cop256 [25] among numerous other postings and publications) herald the moment in the near future when there will be measurable environmental performance standards for historic buildings, comparable to the LEED standard for new construction. At present, the practice is ad hoc, with architects and developers applying newly evolving technologies from other sectors [26]. Government incentives for climate friendly adaptation of existing buildings are likely to fuel this trend.

Social impacts are more intangible and more difficult to quantify and measure. By definition, community collaborations are small in scale and intended to benefit specific social groups (such as genders, racial minorities, immigrants, and disprivileged people). At the other end of the spectrum, programs that benefit society at large (through poverty alleviation, food provision, education, and disease eradication, for example) are broad-based, sweeping in scale, and measured quantitatively rather than qualitatively. While

heritage preservation contributes materially to the quality of life in communities, this impact takes diverse forms by creating safe and nurturing public spaces, providing equitable employment opportunities and job training, and creating a diverse community through equitable housing opportunities. There is no universal methodology to measure these impacts. In its absence, sponsors identify the desired outcomes and community impacts of each project individually, in broad terms.

As the heritage field positions itself to have a meaningful role in sustainable development, it will need to prioritize the issues it addresses. The Climate Heritage Network contributed to defining this territory by stating its objective to focus on "people from vulnerable groups and communities" susceptible to harm and lacking capacity to cope and adapt. It prioritizes:

- Urban communities: to transform urban slums into healthy, clean, and safe cities;
- Rural communities: to equip smallholder farmers to adapt and thrive;
- Coastal communities: to protect homes and businesses against climate shocks

These three priorities, if adopted widely in the field and carefully documented as they are addressed, will create a strong response to the sustainable use of heritage buildings, the objective of SDG target 11.4.

### 4.2. Blended Finance

The term blended finance came into use following the Third International Conference on Financing for Development in July 2015, as a solution to the funding gap foreseen in achieving sustainable development goals, which will require an additional USD 2.5 trillion in private and public financing per year as of 2017 estimates, and an additional USD 13.5 trillion to implement the COP21 Paris climate accord. The heritage field's response to this opportunity and challenge was to enlarge its scope of projects to include direct social and community outcomes, as well as community dialogue in the setting of conservation strategies, to identify points of convergence between sustainability agendas and heritage conservation priorities (as articulated by the Climate Heritage Network above), and to develop a set of tangible references for measuring heritage building compliance in relation to carbon reduction through building retrofit and reuse [26].

CHiFA identified six streams of layered funding with return expectations ranging from zero at the bottom layer to market rates at the top. Concessionary terms and enabling devices at different levels within the capital stack offset senior investment losses and encourage capital flows.

The fundamental level of investment is governmental budgetary allocations, grants, and other nonrecoverable public financial commitments. As owners of most monumental properties in historic areas, governments have the highest legal responsibility and practical interest in economic recovery in distressed communities, even without considering other governmental obligations to deliver health care, education, safety, and other public services that can be delivered through urban regeneration. Laying down a framework of government commitments sets the potential scale and geographical boundaries of an urban regeneration initiative. Municipal funding and the concession of municipally owned buildings figure into this pool of available capital assets.

Intergovernmental agencies and international lenders and donors are the next layer of the capital stack. Funds may come from national or regional incentive programs, from international lenders such as development banks, or from international donors through development agencies and funds held in trust by the United Nations and its affiliates. International financing may be a significant element in an urban regeneration campaign, but this funding is generally concentrated on specific projects under specific conditions. Commitments are made to the recipient government, which provides any requisite guarantees of repayment. Funding from such sources may be significant and highly impactful, but will not complete the picture. Mapping intergovernmental/international funding fills out the second layer.

A third layer of funding is philanthropic contributions from foundations, community funds, individuals, and other private sources. Directed to specific projects, often channeled through local nongovernmental organizations that enrich the growing initiative as civil-society partners, this funding is influential in securing the commitment of other traditional players. Typically, philanthropic donors expect their funds to leverage two to three dollars for every dollar given. Performance expectations may include community, social, and environmental sustainability impacts, public use and enjoyment, and longevity of the cultural resources involved. Philanthropic donors rarely require economic sustainability as a criterion for investment and expect no payback other than specified impact values.

These three layers of public-sector investors and philanthropic donors comprise the traditional range of available funding sources. Securing additional capital requires paving the way to more senior investment. This is the role of the fourth layer—early-stage capital. Early-stage capital is provided by a specific source, such as a revolving fund or recoverable grant, with the goal of creating investability and attracting senior financing. Program-related investments by foundations—an increasingly common way for foundations to deploy capital to address urgent problems and issues—represent a promising new source of early-stage capital for heritage regeneration. The cost of this layer of capital is higher than intergovernmental finance, but lower than market-driven investment, pegged to capital recovery rather than profit. The term of lending or grant recovery is relatively short—up to five years.

Senior investors, with a large potential to provide capital, may use debt or equity instruments, impact, institutional capital, or private capital. Impact investors may be prepared to accept measurable environmental and social outcomes in lieu of market-rate returns. Institutional investors are regulated and require a substantial administrative framework, guarantees from subordinated investors, and a high threshold of capital investment in order to engage. This layer of investor may be attracted by an experienced developer taking an orchestrating role in the overall program, or by well-documented projections of positive economic outcomes. Institutional investors may provide capital in response to a guarantee, or subordinated funding from a local development bank or institutional source. Blended financing from these layered sources provides a holistic framework for sustainable growth while building positive social, environmental, and economic returns over time.

*4.3. Further Insights*

Valuation of heritage as a tangible and intangible asset has been a theme of research and publication for more than two decades. In the last ten years, which have seen the creation of the HUL Recommendations by UNESCO, the Sustainable Development Goals and the UN Habitat New Urban Agenda, the leading institutions in the discussion of cultural valuation, such as the World Bank, the Inter-American Development Bank (IDB), the Getty Conservation Institute (GCI), the International Council on Monuments and Sites (ICOMOS) and the International Center for the Study of the Preservation and Restoration of Cultural Property (ICCROM), updated their research and thinking on cultural heritage as a driver for sustainable development, all with the intention of changing the mindset of the heritage conservation field toward more integration with development issues. This section summarizes some of the key contributions to the growing idea of shared responsibility and economic sustainability for heritage.

In 2012 the World Bank published studies on cultural assets in historic cities as public goods [27], and ICCROM organized a symposium on concrete indicators for measuring heritage performance [28]. The IDB published case studies of heritage-led urban regeneration in ten Latin American cities [29], and the pioneering specialist in heritage economics Donovan Rypkema, together with Caroline Cheong, produced a guide to developing heritage-based public-private partnerships [30]. UNESCO, throughout this period, sponsored continuous dialogue in concert with the preparation, promotion, and application of the HUL Recommendations [31]. Its former Assistant Director General Francesco Bandarin

continued to explore the theme of heritage and uban development in collaboration with R. van Oers [32], M.T. Albert and A. Pereira Roders [33,34].

Europa Nostra, a consortium that advocates for pan-European heritage policy and grassroots initiatives in 2015 published a fundamental value statement for heritage in response to the Sustainable Development goals, intended for wide implementation on a national level [35], and collaborated with Climate Heritage Network hosted by ICOMOS (see note [25]).

The topic of Public Private Partnerships (P3s) as a global vehicle for heritage regeneration was explored by GCI's Susan MacDonald, working with Caroline Cheong in 2015 [36]. A later study by Cristina Boniotti expands the analysis to P3 and P4 partnerships in Italy [37]. To this discussion, S. Labadi and W. Logan added a study of sustainable public management frameworks for urban heritage on an international, national and local level [38].

A recently formed organization, Our World Heritage, sponsored a year-long series of webinars in 2021 on heritage as the focal point of urban development strategy [39]. This work underpins and complements CHiFA's research and publications cited above (see pp. 3–8 and notes [11–19]). The concept of an expanded financial framework for heritage development was first presented in CHiFA's publications, cited above, and at the First International Conference, TMM_CH, 2018 [40].

## 5. Conclusions

CHiFA's research, published in 2021 (Section 2 and notes [25,26]), studied models that successfully brought together financing from a range of sources to effect meaningful positive change by transforming derelict heritage areas into dynamic communities. The methodology presented in Section 3 of this paper further proposes a process for integrating public and private interests and opportunities to create an investment dynamic that can attract blended finance. The discussion in Section 4 synthesizes a group of factors that help to facilitate a successful collaborative initiative—high-level political support, capable local leadership, risk mitigation and investment incentives, measurable impacts, and a transparent management structure.

Launching the regeneration process requires a management entity that can bring together the interests of the different parties, design structures that mitigate risk, and act as an orchestrator of the public intertest translated into investment-ready financial offerings. In many of the cases studied, this entity has been a non-profit organization, but it can also be a private corporation working for public benefit (such as a B-Corp), a social entrepreneur, or a purpose-built public agency.

The practice of heritage conservation has focused on preserving individual architectural assets and their contexts. Its regulatory systems vary from one country to another but all use designation of sites, historic buildings and districts as the framework for legal protection. The individual property-by-property approach has left the field without the large-scale global strategies and priorities. The Climate Heritage Network attempted to identify areas of urgent priority—urban communities, rural communities, coastal communities—in preparation for the 2021 Climate Accords, and these priorities are all appropriate for investment through blended finance.

To address these priorities, the next steps in the process are:

- To catalyze engagement between heritage managers and a broad range of financing sources, by packaging projects in high-priority areas for funding through blended finance;
- To establish metrics for measuring the performance of heritage sites in relation to Sustainable Development Goals;
- To create a vehicle or facility capable of providing catalytic early-stage capital to expedite the development of these steps.

With more investment-ready projects coming to the attention of financial markets, and measurable impacts, it is possible that heritage buildings and areas in communities with significant history will one day be regarded as a nonnegotiable element of urban development planning; that is to say, a defined class of assets with predictable performance

and multiple values that can be measured in relation to delivery of multiple goals. As these steps are taken, a body of knowledge will emerge that more fully, and less anecdotally, documents the case for heritage properties as a class of assets worthy of investment.

**Funding:** This research received no external funding and was supported by the Cultural Heritage Finance Alliance through general support.

**Institutional Review Board Statement:** Not applicable.

**Informed Consent Statement:** Not applicable.

**Data Availability Statement:** Not applicable.

**Acknowledgments:** The author is grateful to Antonia Moropoulou for the opportunity to present this paper in the Special Issue of this journal, and to the Inter-American Development Bank for the opportunity to develop the four-step methodology described in Section 3 of this article. Thanks to Tracy Pickerill and the European Union CLIC program for many insights into heritage financing instruments and enablers. Thanks to my colleagues at the Cultural Heritage Finance Alliance, Norma Barbacci, Laurie Beckelman, Gary Hattem, Derek Moore, Sari Uricheck, and Keith Wright, for continuous collaboration and support in the development of the material published here.

**Conflicts of Interest:** The author declares no conflict of interest. Funders of the Cultural Heritage Finance Alliance had no role in the design of the study; in the collection, analyses, or interpretation of data; in the writing of the manuscript, or in the decision to publish the results.

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
