# Peer review of "A Blended Finance Framework for Heritage-Led Urban Regeneration"

_land, doi:10.3390/land11081154_

Round 1

Reviewer 1 Report

Dear author,

The manuscript is user-friendly and very well-structured. It builds on the strong connection between heritage and sustainable development and the key role adaptive reuse and heritage-led regeneration play in improving the quality of life and in contributing to the green transition globally. The manuscript also provides lessons learned from case-studies (both from developing and developed countries) and draws a very clear list of existing financing instruments. Moreover, the manuscript sets-out a four step process which could be instrumental in articulating a vision and strategy for public-private cooperation in heritage-led regeneration globally. The discussion is sound and strongly informed by evidence-based research.

Author Response

Thank you very much for your comments.

Reviewer 2 Report

The paper is well written and represents a state of the art for future heritage regeneration.

Author Response

Thank you very much for your comments.

Reviewer 3 Report

The article deals with a very relevant and timely topic

However, I have some doubts about the scientific component of the article. I think that this is an article more for global knowledge of concrete and non-extrapolate situations than for the various stakeholders to be able to use in practical terms.

In my opinion, there is essentially a lack of a section with literature on the subject, discussion with an indication of some authors, and, in the end, perhaps introducing the practical implications and contributions of the work.

Author Response

I have revised the conclusion of the article, as well as the abstract, to make it clearer that the blended finance framework can and must be applied to urban heritage regeneration in order to align with sustainable development goals, contribute to closing the sustainability development financing gap, and create sustainable solutions to sustainable and environmental goals.  I have also changed the title of the section references section to Notes and added a general references to the most recent research that has been produced by most established experts who have been working with heritage valuation themes. I am not sure that this is acceptable for this journal. If this revision addresses your concerns, and a separate reference section is not allowed, I will expand the text.  I understand your concern about the scientific approach to the research.  The goal of this article is to present a process that can be widely applied to enhance the scale and impact of heritage projects.  The proof is demonstrated through an analysis of the case studies rather than through empirical testing.

Reviewer 4 Report

This article seeks to propose a hybrid financing paradigm for heritage-driven regeneration. It contributes to the establishment of multinational public-private co-financing programs and partnerships to assist urban heritage regeneration. However, some concerns need to be taken into account.

The beginning of the abstract is unclear and raises fundamental questions. Therefore, I recommend revising them to place greater emphasis on the state of the art and research question. By what approach or process may heritage-led regeneration have good effects on the public realm? Or, is it feasible to achieve a carbon-free environment through heritage? Exists any evidence to support this statement?

It would be helpful if the manuscript explained how effective private investment in heritage sites could be as a catalyst for local economic growth, with good environmental consequences and social value reinforcement complementing these investments in culture?

Although table 1 shows the comparative benefits of various financing methods, it is unclear how these instruments work in the case of a crisis, as there are frequently limited financial resources available for cultural heritage conservation. For example, following the 2008 crisis, Italy reduced its budget for culture, and by 2011, it had been cut in half over the preceding three years, from US$603 million to US$340 million. Please see [https://www.newsweek.com/italys-ancient-monuments-and-cultural-heritage-crumbling-68467]. Consequently, I assume that a critical discussion on the effectiveness of the framework in economic recessions caused by pandemics or natural disasters might improve the accuracy of reasoning.

The third concern is whether the blended finance framework actually incorporates local community priorities. For instance, relevant research demonstrates that despite the fact that city branding (as a hybrid financing paradigm) is an effective instrument for promoting cultural heritage due to its complexity, people and their goals are not sufficiently emphasized in many city branding strategies. Please see [https://doi.org/10.1186/s40410-019-0101-4]. I am curious about how the framework assesses local community requirements in cultural landscapes and incorporates them into investment programs. Are there/are case study(ies) that can be offered as best practices?

While the availability of funds is critical for the long-term maintenance of cultural or rural landscapes, it is not the only factor. For instance, in the case of Cinque Terre in Italy, rural development funds have proven to be effective, and the majority of farmers have applied for them, yet they are insufficient. Even if a farmer restores and maintains his/her dry-stone walls in immaculate shape, hydrogeological issues continue, and his/her cultivations are risked if the terraces of adjoining farms located upstream are not well maintained. Please see [https://doi.org/10.3390/land10020093]. As a result, when building a new financial framework for heritage-led regeneration, it is vital to consider investing in local communities in order to improve resilience thinking in the cultural landscape.

Please update the literature review section to reflect new publications that support the study method.

Overall, I think the manuscript is fascinating research. It should, however, improve its critical discussion.

Good luck

Author Response

Thank you for your review, careful reading and expression of concerns. Concerning community priorities, it is of course a key requirement of project design to address community needs, and this is built into each stage of planning, assessment and identification of funding sources. I have added references accordingly, throughout the manuscript, to aligning with community goals. Concerning the use of this methodology in times of crisis, particularly financial recessions but also other catastrophic events such as natural disaster and conflict, I feel that moments of crisis require another strategy. Building long-term financial partnerships requires a stable governance situation.  I do believe this methodology can be used for situations such as mitigation of coastal erosion and other climate-related impacts that are being addressed systemically, but less so in budgetary crises that require short-term responses, including bridge financing (which is one of the provisions of a revolving fund) rather than the building of inclusive financial strategies.  In response to y our comments about Cinque Terre, this would make an interesting case study, no doubt, and I clearly take your points.  However, the arguments in the article and the case studies are focused on urban situations, which vary somewhat from cultural landscapes because of the public investment in and commitment to  infrastructure.  The article repeatedly makes the point that it is considering urban examples, and I have added the word Urban to the title to make this clearer. 

I have added a section of general references at the end of the article, in addition to the endnotes, to direct the reader to other research that has been published in the last few years.  I am not sure this will be acceptable, and look forward to hearing from the editors. If the article is acceptable in other ways and this needs to be changed to a narrative section, I am prepared to do so. 

Finally, to your point the money is not the only problem, this methodology is of course intended to complement other conservation planning, assessment, and regulatory devices, as well as existing governmental and community efforts. I believe it is widely felt in the field that heritage preservation does not have access from existing sources, and is not likely to have access in the future, to financial resources adequate to catalyze heritage regeneration as a theme of sustainable development. The article focuses on this situation.

I have added a note and anecdotal reference to a carbon reduction strategy developed as part of the EU Clic initiative. This is not a definitive approach, but it points to new directions in the field which are being developed to comply with evolving carbon retrofit requirements in existing buidlings. I believe that it is likely that a global standard will emerge soon, but it is not the goal of this research to propose such a standard.

I have made my best effort to correct typographical errors, and need to

Round 2

Reviewer 3 Report

The authors made an effort to introduce all the suggestions given in the document.

Therefore, I am of the opinion that the article has conditions to be published.

Reviewer 4 Report

The author has satisfactorily answered the comments and concerns highlighted by this reviewer.